



# Non-linear Hydrologic Organization

Allen Hunt[1] Boris Faybishenko[2] Behzad Ghanbarian[3]

[1]Dept. of Physics, Wright State University, Dayton, OH 45435, USA
[2]Energy Geosciences Division, Lawrence Berkeley National Laboratory, University of California, 1 Cyclotron Rd., Berkeley
CA, 94720, USA
[3] Porous Media Research Lab, Department of Geology, Kansas State University, Manhattan KS 66506, USA

*Correspondence to*: Allen Hunt (allen.hunt@wright.edu)

**Abstract.** We revisit three variants of the well-known Stommel diagrams that have been used to summarize knowledge of
characteristic scales in time and space of some important hydrologic phenomena and modified these diagrams focusing on
spatio-temporal scaling analyses of the underlying hydrologic processes. In the present paper we focus on soil formation,
vegetation growth, and drainage network organization. We use existing scaling relationships for vegetation growth and soil
formation, both of which refer to the same fundamental length and time scales defining flow rates at the pore scale, but
different powers of the power-law relating time and space. The principle of a hierarchical organization of optimal subsurface
flow paths could underlie both root lateral spread (rls) of vegetation and drainage basin organization. To assess the
applicability of scaling, and to extend the Stommel diagrams, data for soil depth, vegetation root lateral spread, and drainage
basin length have been accessed. The new data considered here include time scales out to 150Myr that correspond to depths
of up to 240m and horizontal length scales up to 6400km, and probe the limits of drainage basin development in time, depth,
and horizontal extent.

**1 Introduction**

The development of "physically based" and verifiable spatio-temporal scaling relationships is one of the important goals
stated in the publication, "Opportunities in the Hydrologic Sciences," which helped guide the establishment of NSF's
Hydrologic Sciences Program in the Earth Sciences (EAR) Division. Our purpose is to find techniques which may help to
develop and test such scaling relationships.

Hydrogeologists and earth scientists occasionally organize (eco)hydrologic phenomena according to their spatial and
temporal scales so as to locate them in a single figure (National Research Council, 1991; 2001; Bloeschl and Sivapalan,
1995; Loague and Corwin, 2006). Such figures, known as Stommel (1963) diagrams, are illustrative and can, under the right
circumstances, trigger further useful work, such as testing hypotheses regarding appropriate spatio-temporal scaling
relationships and the relevant space and time scales that enter in. Further, the same figures may even be able to clarify basic
tenets about surface compared with subsurface hydrology, and link processes not originally known to be connected, even
those that cross hydrologic boundaries.




In this particular note, we consider a few basic concepts presented in such figures in the light of recent work on the spatial scaling of river networks (Hunt, 2016; 2017a), the spatio-temporal scaling of chemical weathering (Hunt et al. 2014a; Hunt and Ghanbarian, 2016), soil formation (Yu and Hunt, 2017ab; Yu et al. 2017; Egli et al. 2018), and vegetation growth (Hunt,

2017b). Guided by Stommel diagrams, we use these latter relationships to expand our spatial scaling of river networks to a spatio-temporal scaling framework.

Chemical weathering, which is typically limited by downward water flux (infiltration) in the unsaturated zone, is the principle limiting factor of soil development (Egli et al. 2018; Hunt et al. 2021). Thus, soil depth will have a dependence on that downward flux. Vegetation growth is related to transpiration (Hunt et al. 2020) (Figure 1). In each case, the fundamental

network length scale is on the order of a particle (or plant xylem) size. The magnitude of groundwater fluxes, which are often nearly horizontal, is typically in the tens of meters per year (Bloeschl and Sivapalan, 1995), but magnitude of typical flux through the unsaturated zone and groundwater recharge is typically (e.g., Lvovitch, 1973) on the order of 10% - 30% of the total precipitation (ranging from 0.001m yr$^{-1}$ to 10m yr$^{-1}$). Infiltration and transpiration velocities are typically on the order from 0.1m yr$^{-1}$ to 1 m yr$^{-1}$, but extreme climate cases (e.g., the Atacama desert and the New Zealand Alps) may extend

this variability by orders of magnitude.

Our proposed power-law length-time scaling relationship can be represented in the following form (Hunt, 2017b):

$$x = x_0 \left(\frac{t}{t_0}\right)^s \tag{1}$$

In this relationship, $x$ is a distance (for soil the depth; for plant growth--it is the root lateral spread, rls; and for river

networks--it is the length of a river), $t$ is the time, $x_0$ is the network characteristic scale (a pore separation), and the ratio of $x_0/t_0$ is the appropriate pore-scale flow rate, $v_0$. The power $s$ was chosen (Hunt, 2017), for soil development, as the inverse of the percolation backbone dimensionality, $1/D_b = 0.53$, and, for vegetation growth, equal to the inverse of the 2D percolation optimal paths exponent, $1/D_{opt} = 0.83$. These non-linear exponents reflect diminishing connectivity of the network with increasing length scales. It was suggested that the variability in flow rates would be responsible for the most important

variability in actual scaling relationships, though the fundamental length scale could also be of significance. The flow rate variability was assumed (Hunt, 2017b) to be ca. 0.024 m yr$^{-1}$ to 20 m yr$^{-1}$, with the upper limit maybe too high, but which appears to give a good accounting for all three phenomena--soil formation, vegetation growth, and river drainage development. The relevance of Eq. (1) using the same parameters $x_0$ and $t_0$ from Yu et al. (2017ab) and Hunt (2017) will be tested further below by investigating its ability to frame the understanding of the Stommel diagrams.





## 2 Spatial and temporal scales of processes of relevance to hydrology

### 2.1 The Loague and Corwin (2006) Stommel diagram

In the context of our work on spatio-temporal scaling, we modified the Stommel diagram plotted in the chapter "Scale Issues" of the Handbook of Groundwater Engineering by Loague and Corwin (2006), which is shown in Figure 2. We have added the colored strips to indicate important spatio-temporal scaling relationships of Eq. (1). The width of the strips indicates roughly two order of magnitude variability in mean annual flow rates, which is largely a result of water supply and its variability from meteorological driving forces, but also the heterogeneity of soil properties. We also added the icon for mineral dissolution and changed the position of landscape evolution so that it would represent a weathered depth.

The blue strip represents *instantaneous* pore-scale subsurface flow rates within an order of magnitude of 1 μm s$^{-1}$ (about 30 m yr$^{-1}$) (reported in a Stommel diagram of Bloeschl and Sivapalan, 1995, as the range of subsurface flow rates most commonly observed). On a bilogarithmic plot, a process with constant velocity $x = v_0 t$, shows up with only a single relevant parameter (the velocity, $v_0$), which determines the vertical position; the range of likely velocities then generates the range of vertical positions. A flow rate as large as 1μm s$^{-1}$ can be observed in saturated, more nearly horizontal flow, below the water table. However, the largest vertical flow rates in the unsaturated zone are likely to be as much as a factor 3 smaller. Nevertheless, a maximum water flow rate of ca. 0.63 μm s$^{-1}$ = 20m yr$^{-1}$ has proved (Hunt 2017) an excellent predictor of the maximum soil depth over time periods ranging from decades to greater than 10Myr, and is reused here.

In Figure 2, we give the water flow rates as scale independent even though subsurface flow rates are assumed in Bloeschl and Sivapalan (1995) and inferred from observation (Skoien et al., 2003) to increase with scale. The question is of considerable importance, though discussions have not provided unambiguous results. This assumption already underlay the construction of the original figure in Hunt (2017) and is given further basis in Hunt et al. (2014b).

Multiplication of the velocity of reaction products by their molar density yields their rate of reaction, when such reactions are transport-limited. This scaling velocity diminishes as a negative power of the time, according to the time derivative of Eq. (1) (Ghanbarian-Alavijeh et al. 2012; Hunt et al. 2014a; Hunt and Ghanbarian, 2016). Proportionality of the reaction rate to the soil formation rate allows for a prediction of the soil depth using Eq. (1) when erosion rates are negligible (for times less than required to reach steady-state, Yu and Hunt, 2017a). Thus, the simple scaling relationship that defines the solute transport distance (Eq. (1)) for advective solute transport in three-dimensional heterogeneous porous media (Ghanbarian and Hunt, 2012) together with the same range of flow rates yields the brown strip.). In this Stommel figure, we have used the value of the instantaneous flow rate compatibly with both Loague and Corwin (2006) and Bloeschl and Sivapalan (1995). But, in order to make individual predictions of soil depth as a function of time (Yu and Hunt, 2017ab; Egli et al., 2018; Yu et al., 2019), the flow rate for each field site where soil depths are measured must be chosen as the annual mean value for the vertical flow in the unsaturated zone, expressed as the quotient of P - ET and the porosity (P = precipitation, ET = evapotranspiration), which is typically an order of magnitude (or more) smaller than 20 m yr$^{-1}$. The vertical placement of the brown strip is chosen to make the ranges of solute and flow velocities equal (brown and blue strip widths equal) at the length



scale corresponding to a typical silt-sized grain of about 30 μm (and time scale of around 30 seconds). Thus, the reduction of solute velocity with increasing scale sets on at the fundamental network scale of a single grain. Note that this band matches

well with the icons of "chemical weathering," "soil depth," and "landscape evolution," as well as reasonably well with "soil structure" and "mineral dissolution." This suggests the interpretation, confirmed in a number of references (Yu and Hunt, 2017ab; Yu et al., 2017, 2019; Egli et al., 2018), that use of the solute velocity in advective flow to predict chemical weathering and soil formation rates is accurate on the average to within 20% (see, particularly, Egli et al. 2018 for verifications).

The green strip in Figure 2 is defined through the optimal paths exponent of percolation theory in two dimensions and represents equally total transpiration as well as root lateral spread (RLS) as a function of time (Hunt et al. 2020b). Optimal paths are defined by the minimal flow resistance. For periods up to a century, RLS are nearly the same as tree height (Hunt and Manzoni, 2015, based on data from Phillips et al. (2014; 2015), Kalliokoski et al. (2008), and Stone and Kalisz (1991)). The vertical position of this band in Figure 2 is also chosen so that the variability of the root tip extension rate at the time

scale of about 30 seconds is equal to that of the water flow rate (blue strip) in the medium. The accompanying interpretation is that roots growing through heterogeneous soils tend to follow paths with lowest cumulative resistance. The access to nutrients near the surface (delivering the 2D topology assumed) provided by this growth pattern is essential for plant growth. Note that the green band appears to be consistent with the icons "grass" and "leaf canopy," while it is also in accord with "vegetated buffer strip." It is reasonable that "soil moisture" extends across the flow and vegetation growth bands, since the

former position on the diagram represents the rate at which moisture is delivered to the soil from precipitation and the latter represents the dominant rate at which it is taken up by transpiration. The mathematical predictions for vegetation growth have been verified separately (Hunt, 2017b; Hunt et al., 2020ab).

While a number of icons fall into the positions expected if such phenomena are indeed governed by the processes suggested, there are a few that fall notably outside the predicted ranges. "Crops" fall on (or below) the blue strip. The relatively minor

limitation on crop growth to a large *flow* rate is provided by ample watering and fertilization, meaning that root growth in a nutrient/water search is not a limiting factor; indeed crop growth is mostly linear in time (Hunt, 2017b). "Forest" falls also below the blue strip, but for a different reason. Reflection on the possible relevance of a length scale of 100m or more in a year indicates that this length does not refer to RLS or the height of individual trees, but more likely to the horizontal spread of forests into newly available habitats. This spread can be mediated by atmospheric processes, such as seed dispersal and,

thus, is not subsurface limited. As expected, "weathering rind" thicknesses are less than "soil depths," since, e.g., subsurface clasts used for measuring weathering rind thicknesses have much lower hydraulic conductivity values than the soil (Sak, 2021), and the amount of water entering the weathering rind is reduced by approximately the contrast in hydraulic conductivity values. Finally, the position of "soil texture" at around 30 μm after about one thousand years represents a typical grain size in the silt range, compatible with the fundamental scale of the network, though it could also be located at

scales of centimeters, which defines a typical length scale on which heterogeneity of soil texture is defined.





An important principle revealed in Figure 2 is its organization of surface and subsurface phenomena. The fundamental subsurface flow rate is not assumed to change with scale. Note that all other subsurface processes fall to the right of the flow line (consequently, as will be seen from Figures 3 and 4, drainage basin development is fundamentally a subsurface process), and all atmospheric processes to the left. Thus, the other subsurface velocities indicated on the diagram are less than the
subsurface flow rate and all surface velocities are greater. Further, the radial pattern of icons diverging from the upper left suggests that subsurface process rates, except that of water flow, tend to decrease with increasing scale, while flow rates associated with the atmosphere tend to increase with scale until limits of planetary size are approached. The decrease in subsurface velocities with increasing scale is associated with the heterogeneous networks over which the processes evolve and the diminution in connectivity with increasing scale. The increase in velocity of atmospheric processes with increasing
spatial scale is a sign of their foundation in non-linear dynamics, and is more easily seen in the inverse view where momentum transfer to smaller scales through turbulence is terminated by frictional interaction with heterogeneities on the Earth's surface. This produces a decrease in fluid flow rates with decreasing length scales approaching surfaces. Fluvial processes along the surface appear consistent with a velocity that increases slightly with scale and is certainly larger than the subsurface flow rate. According to Skoien et al. (2003), indeed, the appropriate dependence is represented by $t = A\,x^{\,0.9}$,
which is in accord with the diagram. This result is generally consistent with a non-linear flow phenomenon interacting with surface heterogeneity.

An increase in groundwater flow rate would be expected to increase advective flow limited processes in the subsurface as well. Without such an increase, increased carbon sequestration is likely to be limited, except for mitigation of the loss of sequestration due to (soil) ecosystem damage. In order to speed up the water cycle, groundwater flow rates would have to be
increased, or else the limiting effect of groundwater flow speeds would need to be bypassed. Where water flow rates are flux-limited, an increase in precipitation will be reflected in groundwater flow, but where the hydraulic conductivity is too small to accommodate increased fluxes, the increased hydrologic fluxes will mostly tend to increase surface run-off. But bypassing groundwater flow paths through higher velocity surface paths (due to land-use changes) will only allow an acceleration of the water cycle unaccompanied by effects on the carbon cycle, which are mostly associated with chemical
weathering and plant growth.

## 2.2 The NRC (1991) Stommel diagram

The publication, "Opportunities in the Hydrologic Sciences," known as the "Blue Book," (National Research Council, 1991) was fundamental to the foundation of the Hydrologic Sciences program at NSF. Figure 3, presenting the modified Figure 2.9
(on page 59) of Garrison Sposito from the book "Opportunities in the Hydrologic Sciences" illustrates ranges of hydrologic process scales. We should point out that the original Figure 2.9 was reused as Figure 2.2 in the 2001 report "Basic Research Opportunities in Earth Science," which figured in the organization of the CZO observatories. In the 2001 publication, the





axes were distorted somewhat, such that equal intervals do not correspond to equal ratios of time. We reproduce the original figure here as Figure 3 overlain by predictions derived from subsurface flow and vegetation growth (Hunt, 2017b).

In Figure 3, the length scale associated with "soil depth" icon clearly does not refer to depth, but rather, to horizontal length scales; otherwise we could have a 1000km deep soil in about 10,000 years.

Consider next the three icons, "shallow ground water circulation," "channel network formation," and "development of major river basins" that form an approximate line. What could that line represent? Hunt (2017a) compared percolation theoretical expression for tortuosity with implications for stream sinuosity of Hack's (1957) law relating stream length to drainage basin

area,

$$L = C(A)^p \hspace{6cm} (2)$$

When miles are used as units of length, $C = 1.4$ (Rigon et al. 1996), but for kilometers, $C$ is $(1.4)(1.6) = 2.24$. The values of the exponent $p$ are found to be between 0.57 and 0.6 (Maritan et al., 1996, though Rigon et al., (1996) argue for 0.6), which

produce stream sinuosity (ratio of stream to basin length) exponents exactly twice as large (between 1.14 and 1.2) because the basin area and the basin (not river) length relationship is Euclidean. The tortuosity exponents of percolation theory (1.13 for random networks, and 1.21, for strongly heterogeneous networks, Hunt and Sahimi, 2017) thus described rather precisely the range of observed stream sinuosity. Our question here is whether there is a link through these exponents to the fundamental pore network scale? And does this link then extend to a spatio-temporal scaling function (Eq. 1)?

If we use the geometric mean annual flow rate from Figure 4 (2m yr⁻¹) and the same fundamental length scale of 30μm together with either the simple tortuosity exponent, 1.13 (the green line), or the optimal paths exponent, 1.21 (the brown line), the three considered icons relating to shallow groundwater circulation and drainage basin development are covered nicely. We wish to investigate this possibility below. First we check whether evidence exists that groundwater flow can be important to drainage development.

Petroff et al. (2013) related river drainage development to subsurface flow patterns by looking at flow convergence in areas with insignificant relief, such as Florida. The verified relationships extended to bifurcation angles. Brocard et al. (2011) and Yanites et al. (2013) have also noted a potential role of subsurface flow in stream capture in that groundwater piracy may accelerate mass wasting and erosion of an interior divide. Laity and Malin (1986) and Baker et al. (1990) demonstrate that groundwater sapping (flow convergence, but also seepage induced chemical erosion) controls the rate of headward migration

of drainages over long periods of time, as well as the architecture of amphitheaters. However, tectonic processes also contribute to changes in subsurface potentiometric gradients. Thus, important roles of subsurface flow in drainage basin development have already been recognized, and the present implication merely extends the apparent relevance to other properties as well as larger length and time scales.

Given the possible influence of subsurface processes on drainage basin development, the seeming correspondence of the

icons of the Stommel diagram in Figure 3 that represent river basin development, and the apparent coincidence of plant root development and river path development along optimal paths in a highly heterogeneous environment, we propose now





investigating the spatio-temporal scaling of drainage basins in terms of Eq. (1), which was found to predict the range of root lateral extent values in the subsurface over time scales up to 100kyr. Our results are shown in Figure 4.

Before discussing the interpretation of Figure 4, we discuss the variability in the theoretical predictions. For plant roots, this

variability is represented in terms of variable flow rates, while the variability of the fundamental network scale is smaller and is neglected. For drainage basin development, we will assume almost the same variability in the parameters that define the fundamental subsurface variability, namely a typical network scale of 30μm as well as the variability in flow rates (between 0.25m yr$^{-1}$ and 25m yr$^{-1}$).

These results for drainage reorganization can be quantified in terms of length scale (full length of captured river or tributary)

and time. The latter is measured starting with (typically) a trigger event from tectonics (sometimes a previous stream capture) and ending with the integration of the drainage. Such length and time scales, while not arbitrary, are also not unambiguous, and considerable uncertainty is present. Frequently, neither the date of the initiation of the drainage basin change nor that of its equilibration is known accurately. For dates that were given in geologic terms, such as early Pleistocene, we used the standard mean of the stated interval. Dates of tectonic triggers are more broadly defined, while

discrete steps of drainage integration (or disintegration) due to steam captures are sometimes more narrowly defined. Sometimes the authors did not give river lengths, but only drainage basin areas. In such cases, Eq. (2), Hack's law, could be applied to estimate a river length.

*Multiple events: Stream captures*

Here, four cases are discussed in which two stream capture events yield two experimental points. The capture of the upper

Amazon by the lower Amazon (total length 6400km) occurred at about 10 Ma ca. at least 56 Myr since tectonics began to impact an established late Cretaceous flow in nearly the opposite direction (Hoorn et al., 1996; Filgueiredo et al., 2009; Albert et al., 2018). In response to this capture event which increased trunk flow, the Amazon basin began to expand northward (Hoorn et al., 1995) with the result the Rio Casiquiare a tributary of the Rio Negro, itself a tributary of the Amazon, is currently capturing the Orinoco River. The Casiquiare (356km) and upper Orinoco (ca. 480kn) combined are

about 840km long, and the time required for integration is on the order of 10Myr. Information on related drainage reversals was inadequate for the present purposes. A second such sequence of events started with the capture of the upper Colorado (overall length 2300km) by the lower Colorado about 11 Myr after the split of the lower Great Basin (17Ma, Young and Spamer, 2001) at about 6Ma. Partly in response to that capture, a 290 km long tributary of the Colorado (the Gunnison) changed course around 1Ma (Aslan et al., 2014) and abandoned Unaweep Canyon. A third such sequence of stream captures

in the Appalachian Mountains is discussed in Prince et al. (2011), with a 225km$^2$ basin connected to an existing stream in 1-2Myr, and a 7000km$^2$ basin expected to be captured in a total of 5Myr. For these two examples, Hack's law (initially derived for this same physiographic province) had to be used to convert basin area to stream length. Yanites et al. (2013) discuss in their Figure 2 two steps of the reorganization of the upper Rhine River: a 1.3Myr duration capture of the Aare/Doubs (ca. 450km), and then a 1.2 Myr capture of the Rhine above Waldshut (300km).



In two cases, because dates of original triggering events are not available sequential captures yield only a single data point. Along the course of the Meuse (Benaichouche et al. 2016), state "the oldest piracy [leaving the Bar river in an oversized valley] corresponds to the capture of the *Haute Aisne* by a tributary of the Oise River. The time of the piracy was ~ 900 ky ago. Later, at ~ 250 ky [and about 150km upstream] the Ornain, and Saulx Rivers were captured by the Marne River." The length scale of 150km thus corresponds to the time scale of 650 kyr. Each capture shortened the original Meuse River

drainage. In Guatemala Brocard et al (2017) summarize their findings: "these diversions occurred during the last 600 ky with the following succession of events: 1) capture of the Cahabón River near Santa Barbara at ~500 ky; 2) capture of the Cahabón River near Purulhá [ca. 50km downstream] sometime later (240-450 ky ago) [mean of 345 ky ago]. The case for using the particular time and space intervals here is not as clear as with the Meuse (Bernaichouche, 2016), for which the succeeding captures were developed from different streams whose drainages were subparallel and integrated. In the case of

the Cahabón River, two unrelated drainages were involved in the captures. Both were considered from the perspective of the disaggregation of a river drainage.

*Multiple events: Sill overtopping*

The initiation of the Mojave River drainage system in California is considered to be about 3.5Ma and it became integrated to a length of 200km by about 20ka. The Mojave River was dammed at Afton for 160ky through pluvial climates before it

finally breached the sill and advanced to the Soda Lake about 40km downstream (Reheis et al. 2012). Less than 10kyr later, the Mojave River arrived in Dumont lake, 50km further, and likely reached Death Valley, nearly 150km further downstream, but Holocene drying interrupted integration of this drainage system (Enzel et al. 2013). If the Pleistocene pluvial period had continued, its full integration to the distance of Death Valley would have been possible in the future.

Per Larson et al. (2020) "A ca. 2.5 Ma age for the initiation of top-down integration of the Verde River from the upper Verde

Valley into what are now downstream basins is consistent with the presence of a 3.3 Ma volcanic tephra…" "The basins depicted here were formerly endorheic, but integrated within the last ~2.2–2.8 Ma. The integration of these basins resulted in the modern through-flowing drainage networks of the (320km) Salt, (272km) Verde, and Gila Rivers of central Arizona." The same time frame was implicitly extended to the Salt River, but not necessarily to the Gila River, of which the Salt and the Verde are both tributaries.

*Parallel capture events. Durations of events assumed similar.*

Pastor et al. (2012) examined 14 (parallel, but not simultaneous) stream capture events in the region south of the High Atlas in Morocco which occurred over roughly 100kyr time intervals. The lengths of the drainages involved ranged mostly from 20 to 40km, from which we extract two data points. Isotopic evidence from the Indus delta indicates the capture of four Punjabi rivers at about 5Ma (Clift and Bluzstajn, 2005). The authors state, "exhumation histories for the western Greater

Himalayas shows that these mountains were in existence before 20 Myr ago," indicating a time period of approximately 15Myr for their capture. The rivers are between 500 and 1400km in length and are named, Ravi, Sutlej, Chennab, and Jehlum.

*Single events.*



Stokes and Mather (2003) addressed the connections of the east-flowing 100km long Almanzora River across the Sierra
Almagro. Quoting the authors: "Emergence and establishment of continental conditions within the basins to the north and
south of the Sierra Almagro during Salmerón Formation time (Late Pliocene–Early Pleistocene [midpoint 2.56Ma]). The
connection probably took place at some point between the Plio-Pleistocene and Pleistocene stages of drainage evolution
(Early–?Mid-Pleistocene)" (1.43 Ma). The time difference is roughly 1.1 Myr. The 4180km long Niger river drainage began
to form between 45Ma and 40Ma when the inland part of its drainage began to emerge from a shallow sea and connection of
the upper and lower reaches was established between 34Ma and 29Ma (Chardon et al. 2016), yielding 11 Myr. Struth et al.
(2020) discuss the reorganization of the 170km Suarez River basin, including its piracy of additional smaller basins along its
east side, over a 405kyr period from its capture by the Magdalena. However, specific distances for the smaller events were
not possible to extract. Goudie (2000) describes the reversal of the eastern 220km long portion of the proto-Katonga river
due to the westward migration of its headwaters in the "swamp divide" over the ca. 400,000 years (Johnson et al. 2000) since
the formation of Lake Victoria through downwarping. Fan et al. (2018) demonstrate the reorganization of the Daotang basin
within 80kyr of the capture of Yihe River by the Chaiwen, adding 25km$^2$ to the Yihe River drainage. Hack's law was used to
generate a distance from the basin area.

The Yellow River reorganization included a possible combined capture at the edge of the Tibetan Plateau near the Gonghe
Basin and top-down organization involving filling of basins from upstream. "Late Miocene to early Pleistocene sediment
accumulated almost continuously in basins until the integration of the Yellow River at 1.8–1.7 Ma (Zhang et al. 2014). The
reorganization was likely triggered due to initiation of faulting in the range 10-11 Ma (Meng et al. 2020). The length of the
Yellow River above the Gonghe Basin is about 1300km. The approximately 100km long paleo-Daotanghe (which means
reversed river in Chinese) was cut-off from the Yellow River 500ky later by isostatic adjustment to the Yellow River
triggered denudation, with its modern remnant of 40+km flowing in the opposite direction to a closed basin (Zhang et al.
280    2014).

*Disputed time scales*

In their Figures 6abc, Wang et al. (2018) demonstrate a 50 Myr duration of the reorganization of the 6400km long Yangtze
across the Jianghan Basin, including flow reversal for the upper half. It should, however, also be mentioned that Su et al.
(2019) place the capture of the middle Yangtze from the Red River by the lower Yangtze at 5Ma, 45 million years after its
alleged integration by 50Ma (Wang et al. 2018). We used the longer interval.

The integration of the Blue Nile drainage with its length of 1450km is relatively well constrained from the emplacement of
the volcanics between 30Ma and 29Ma that capped the Ethiopian plateau to captures at 10Ma and at 5Ma that led to 2-fold
and 5-fold increases, respectively, in its sediment load (Giachetta and Willett, 2018) (However, Fielding et al. 2018,
conducted a detailed geochemical investigation of the Nile delta to greater depths than previously, producing a more nuanced
interpretation including much earlier trans-Saharan connections). We used the longer interval, since the more detailed
evidence of the later study does not controvert the specific conclusions of the earlier one.



Suhail et al. (2020) suggest the integration of 500km of the upper Dadu river in the last 3.8Myr, although since Yang et al. (2020) suggest that the reorganization occurred within 1.4Myr, or even 0.6Myr, the point at 1.4Myr is also plotted together
with the suggested 3.8Myr.

Comparison of the reported results with predicted lengths from Eq. (1) using the given time scales together with identical pore-scale parameters as for vegetation is given in Table 1. Plotting the predicted length scale from Eq. (1) against the observed value, with the condition that the regression pass through the origin, yields an average discrepancy of 8% and an $R^2$ value of 0.78.

A search for subsurface flow rate scaling information turned up an excellent suite of 9 measured basin residence times for deep groundwater as a function of length scale as measured by $^4$He fluxes from South America (Aggarwal et al. 2014). That study does not support a scale-independent flow rate, rather, it supports a *decreasing* flow rate with increasing length scale. In fact, the observations are in almost exact agreement with Eq. (1) for optimal path flow as the measured exponent is 0.824, less than 0.4% different from $1/D_{opt}$, while the numerical constant implies a flow rate of 35myr$^{-1}$ at a length scale of 30μm,
only ca. 50% larger than the assumption in Hunt (2017b) and reapplied here (25m yr$^{-1}$). Furthermore, the $R^2$ value is 0.84. Adding these values to the analysis of the suite of river lengths investigated above leaves the discrepancy unaltered at 8% and changes $R^2$ only from 0.78 to 0.80.

## 3 Uncertainties

In addition to uncertainties in measurements and model parameters common to all the processes discussed, drainage basin
evolution adds the difficulty of dating and measuring a complex process operating in three spatial dimensions, for which much of the immediate evidence is destroyed over time. A motif that is repeated in the literature discussions referenced above is a kind of dichotomy between the perspectives of 1) a new establishment of a drainage and 2) the evolution of an existing drainage, which may in many ways resemble its present form. Such evolution will typically contain events of tectonic and climatic origins, both leaving some imprint on the hydrology, whose evolution may, at times, be discrete and
datable. Thus, what from a static perspective looks like a sudden drainage basin connection, appears, in a more integrated perspective, often to be geographic shifts of an existing drainage through smaller events. Geomorphologists, hydrologists, sedimentologists, geochemists, and geodynamicists have differing perspectives, but in fact drainages may have existed a long time already, affecting both connection times and connection length scales, and increasing the number and complexity of the shifts. A wide range of additional uncertainties in interpretation exists, which will not be discussed in detail here, but it
is clearly to be expected that the dates and spatial scales of fluvial evolution will themselves evolve in the next decades. If inferred connectivity time scales turn out to be much smaller than generally accessed here, that evidence will favor a greater role of top-down drainage evolution (Hilgendorf et al. 2020), by basin overtopping, than by headward erosion. The former proceeds more importantly over the surface than through subsurface hydrologic connections, which influence headward erosion. If the overtopping interpretation is correct, we might expect that the evolution of drainage basins would ultimately





more nearly follow the results for fluvial process icons on the Stommel diagram of Figure 2.Although the cases quoted here
       from the arid southwest fit well with the time scales for headward erosion from more humid climates, large changes in
       climate from the Pleistocene glacial episodes to the (short) interglacials may complicate interpretation that subsurface flow
       rates may be significant even when organization of strings of endorheic basins proceeds by sill overtopping.

## 4 Limits

Consider Figure 4 once more. The maximum time scale found for soil depths is (150Myr) and river basin organization (100
       Myr) correspond to mid to late Mesozoic, if continuous development to the present is appropriate. The predicted maximum
       drainage basin length at 150 Myr is ca. 10,000 km, not very different from the size of a supercontinent, and not so different
       from the time scales proposed for reorganization of the ca. 6400km long Amazon and Yangtze (50Myr – 100Myr). The
       middle of the predicted range of soil depths is a couple of hundred meters, also corresponding to observation of the vertical
extent of the critical zone. Notably, 150 Ma corresponds also to the breakup of Pangaea (Cawood and Hawkesworth, 2015).
       As stated by Gupta (2007) "There is thus a link between the evolution of large river systems with the Wilson Cycle – the
       creation and destruction of supercontinents, such as Rodinia and Pangaea." Given the geologic reworking of Earth's surface
       (and reorganization of mountain ranges as well as oceanic boundaries) on a Wilson cycle time scale, we expect a significant
       drop-off in the availability of relevant and uninterrupted hydrologic and soil depth data at or near the time scale since the
breakup of Pangaea. This discussion justifies the "limits" in time and space put on Figure 4.
       We can use the predictions that underlie Figure 4 to generate an exact (though not necessarily accurate) prediction of the
       architecture of basins at 150 Myr, namely a maximum weathered profile depth and a maximum linear extent (Table 2) We
       apply the maximum and minimum flow rates from Figure 4 as well as the geometric mean. Use of the same flow rates for
       soil formation (infiltration fluxes) and vegetation growth (transpiration fluxes) is reasonable, even though one can expect the
latter, on average, to be about a factor of two larger (Hunt et al. 2020a). Some uncertainties may arise, however, from using
       the same range of flow rates for all three phenomena simultaneously. In spite of these complications, which may lead to a
       modest overestimate of soil depth, we simply use the same velocities and the same fundamental length scale that led to the
       predictions of Figure 4. Clearly, these values are on the right order of magnitude. The estimations may help constrain
       sediment and carbon fluxes over geologic time scales.
Consider the dependence of maximum drainage basin length on flow rates. The longest interconnected drainage under arid to
       semi-arid conditions (P on the order of 10cm yr⁻¹), is predicted to be only 248km. Although this may be a bit extreme, it is
       also clear that in truly arid regions of under 10cm yr⁻¹ (central Australia, Saudi Arabia, and the Sahara) development of
       connected long drainages is uncommon. As pointed out by Maxwell et al. (2016), subsurface flow rates derived from
       residence times are proportional to the difference of precipitation and evapotranspiration. Thus, while wet climates can
generate a drainage basin of continental size (over 10,000 km), in dry climates, lack of connectivity of drainage basins likely
       prevents the organization of such large basins. Further, drying climates will disconnect and inactivate large drainages, such



as the drying of the Sahara since the end of the Miocene (Goudie, 2005) or even, in the Holocene, the incipient Pleistocene Mojave River connections (Reheis et al. 2012; Enzel et al. 2013).

As for specific depths of weathered profiles dating back to Mesozoic times, values of about 130m have been reported in
Germany for near 150 Myr (Felix-Henningsen 2018), of 150m near Belgium (Felix-Henningsen 1990), 120m in Australia (though not continuously forming over the entire interval, Gardner, 1957), over 200m in southwestern England (Migon and Bergstrom, 2001), in excess of 100m South America (Giannini et al. 2017; Ruckmick, 1963), with the deepest at 240m (Ruckmick, 1963). Tardy and Roquin (1992) state: "Over most of the areas of the Brazilian and the African shields, a very thick lateritic mantle has been continuously formed over more than 100 Myr. Laterites are widespread in Australia, India,
Burma, Brazil and in intertropical Africa. In these regions, bedrocks are generally weathered to depths of over 10 to 150 m." Other references to very old weathered profiles are given in Yu and Hunt (2017a and Hunt et al. 2021).

Note that in Figure 4, erosion is not considered, though it has been addressed in Yu and Hunt (2017ab) and Egli et al. (2018) in greater detail. Its full treatment is more complicated than given here. However, it is known that erosion rates as small as $1m\,Myr^{-1}$ occur only in arid continental interiors (Bierman and Nichols, 2004), such as Australia, where the precipitation can
be as low as $0.04m\,yr^{-1}$. In wetter climates, even a denudation rate of $5m\,Myr^{-1}$ would add up to 750m over 150 Myr time scales, which means that our estimates of weathering profile depths are probably not particularly high.

Interaction with spatio-temporal scaling associated with tectonic drivers may introduce an important cross-over in the variability of drainage basins with scale. According to Roberts (2019), "At large scales ($\gtrsim$10 km,$\gtrsim$1 Ma) drainage networks appear to have a synchronized response to uplift and erosional processes. At smaller scales erosion generates complex
landforms. Most power and commonalities reside at long wavelengths and timescales (>100 km,>1 Ma). The presence of commonalities between drainage networks is expected for systems in which large signals (e.g., uplift) are forced through random media (e.g., lithology, biota)." Such an increase in commonality may reflect an interaction between relief producing (tectonic) and relief shaping (drainage basin organization) processes, as Roberts (2019) says, "a synchronized response." Consider Figure 5, where only the river drainage data are retained. Typical horizontal tectonic velocities are ca. $2 - 3cm\,yr^{-1}$
(though oceanic plates often reach $6cm\,yr^{-1}$), a time independent velocity. The reference to the key length and time scales, 100km and 1 Ma, generates an upper limit on horizontal tectonic velocities of ca. $10cm\,yr^{-1}$. In Figure 5, tectonic rates of $3.5cm\,yr^{-1}$ and $10cm\,yr^{-1}$ are given, together with the groundwater flow rate and its associated optimal paths scaling function from Eq. (1) using the same parameters as in Figure 4, but including a range of flow rates of one order of magnitude. Because the slopes of the scaling relationships are so similar (1 for flow, 0.83 for optimal paths), the relatively small
variability in parameters can change the intersection of these lines from 1 kyr to 100 Myr; The maximum tectonic rate and the *given* $25m\,yr^{-1}$ flow rate, for example, meet in the upper corner of the diagram.

The data found for drainage basin reorganization commence at about 100kyr and 10km. Visual examination of Figure 5 may suggest that drainage basin development at longer time scales is confined between the tectonic and the optimal paths scaling functions. A related implication seems to be that, when the optimal paths function falls below the tectonic relationship, no





time scale will suffice for equilibration of hydrology to tectonic changes that plot up beyond that time scale. This restriction may be especially important in arid regions.

Use of a groundwater flow rate of 10m yr$^{-1}$, just over a factor 2 smaller than applied, leads to equality of tectonic and river spatial scales of 100km at exactly 1Ma. If we had used the fundamental groundwater flow rate extracted from Aggarwal et al. (2012) of 35m yr$^{-1}$, virtually all the drainage basin data would lie between the optimal paths scaling and the typical

tectonic rates. Finally, for intermediate values of these fundamental velocity parameters, there is a large range of time scales starting near 1Myr and extending to the end of the diagram, in which the time and length scales of Eq. (1) and the optimal paths scaling match up with those from tectonics. This approximate equality may help explain Roberts' (2019) "synchronized response" from above.

In his opening sentences, Roberts (2019) states: "Evolution of Earth's surface is a result of geological and geomorphological

processes operating at a range of spatial and temporal scales. Therefore, a framework that unifies short wavelength (≲10 km) process-orientated landscape evolution models with longer wavelength observations and phenomenological approaches is attractive." One should then conclude that a theoretical treatment of optimal flow paths unifying observed space and time scales at the pore scale with observed vegetation growth at scales up to 100kyr and 10km, and with continental wavelength observations at up to the Wilson cycle for rivers, is interesting as well. That the solute transport scaling unifies the pore scale

and the weathered depths over the same range in time scales is probably of additional interest.

**5 Summary**

We have introduced predictive scaling equations in subsurface hydrology that incorporate fundamental and universal spatio-temporal scaling relationships from percolation theory populated by parameters that represent the fundamental nature of the porous medium as a network and flow rates as given by hydrologic variables. These individual relationships were confirmed

separately by comparison with a great deal of data that examined their individual predictions for soil depths and vegetation growth. Here, the implications of these (non-linear) scaling equations to the general organization of hydrologic processes with their roots in subsurface flow are considered and some processes not necessarily recognized as being limited by subsurface flow also identified, such as river drainage organization. In the context of a wider understanding of hydrology, it is important to keep in mind that non-linear spatio-temporal scaling does not necessarily imply non-linear dynamics. In

particular, the contrast between the dominant effects of disorder (heterogeneity) in the subsurface and disorder in the atmosphere (non-linearity) leads to fundamentally distinct non-linearities; in the former velocities tend to be non-increasing functions of length scale and in the latter, non-decreasing. Further, in the former, the role of the reproducibility and predictability of experiments is emphasized, in the latter the irreproducibility and sensitivity to initial conditions is paramount. This is not to say that heterogeneity is not expressed in the atmosphere, nor that chaos is absent in the subsurface.

In fact, a kind of slow chaotic process, that of plate tectonics, operates at more or less constant speeds, but produces events, such as rift formation and plate collisions, that appear as sequences of discrete events in the geologic record. Each major





event, such as the aggregation of supercontinents, or their breakup, generates a cascade of related events, which lead to drainage reorganizations.

An important first question is, over how wide a range of length scales can our predicted scaling relationships hold? The depths of soils (or deep tropical weathering) are available for time scales up to about 150 Myr, with the deepest exceeding 200 m. Widespread depths exceeding 100 m are cited. The time for river basin organization to reach continental scales, on the order of 10,000 km, also appears to be about 150 Myr. Both of these numbers are in accord with the prediction of the scaling relationships accessed. That the two already existing scaling relationships predict reasonably accurately the largest horizontal and vertical scales of what is essentially the critical zone over a geologic time scale equal to that since the separation of Pangaea, suggests anew the fundamental relevance of the Wilson tectonic cycle in constraining our knowledge of hydrogeomorphic evolution. The coincidental near equivalence of the optimal paths scaling function and its slow decline of velocity over time with the velocities associated with tectonics at time scales from approximately 1Myr to 100Myr seems likely to form the basis for the commonalities of drainage basin evolution displayed on these time scales, as well as the greater variability at shorter time scales, particularly on smaller regions of continental land mass. It also suggests a greater relevance to continental scale processes of flow and transport paths traced all the way back to pore scales than hitherto typically assumed. Subsurface flow paths and rates appear to be critical to drainage basin development.

Our concluding points follow:

- Approaching spatio-temporal scaling diagrams with a network theory of flow and transport allows spatial and temporal scales of seemingly different processes to be unified in terms of concepts, theories, and flow rates.
- The scaling relationships can hold over time scales from seconds to hundreds of millions of years.
- Hydrologic observations appear to be constrained significantly by the Wilson tectonic cycle of supercontinent origin and breakup (most recently, 150 Myr).
- Figure 4 shows that the subsurface water flow rate is a critical control of hydrologic processes, limiting both managed (crops) and natural plant growth, as well as soil formation, and, in this preliminary investigation, appearing to limit river drainage development.
- Figure 5 shows the near coincidence of tectonic and optimal paths length scales over a wide range of time scales beyond 1Ma, perhaps providing a mechanism for more universal drainage basin characteristics in this range of time scales.
- Relevant processes for which spatio-temporal scaling reveals characteristic velocities slower than the subsurface flow velocity appear also to have velocities that diminish with length scales. As such, we suggest that their predominant influence is from heterogeneity.
- Processes for which spatio-temporal scaling reveals characteristic velocities faster than the subsurface flow velocity appear to be associated with velocities that increase with length scales. We suggest that these processes are






primarily influenced by non-linear dynamics. Such velocities can reach a maximum on spatial scales of continental separations.

- Clearly, surface flow is influenced by a wide range of surface heterogeneities as well as non-linear dynamics., Fluvial processes (Figure 1) appear to exhibit scaling with a characteristic velocity slightly higher than subsurface flow rates and with a velocity that appears to increase slightly with increasing scale. We suggest that this result implies a somewhat greater relevance of non-linear flow equations than heterogeneity to the dynamics of surface


flow, in spite of the apparent dominance of heterogeneity in the subsurface to the evolution of the shape and position of the flow paths and the architecture of drainage basins. Such a result is also in accord with suggestions that river drainage reorganization by overtopping of barriers to flow should occur more rapidly than by headward erosion; however, the apparent success in predicting time scales for drainage reorganization using the subsurface optimal flow paths scaling relationship, suggests an important role of subsurface flow in some drainage


reorganization events, even when the apparent hydrologic controls relate to surface features.

Finally, we hazard a prediction. Willett et al. (2014) use drainage basin disequilibrium calculations to analyse the capture of the Apalachicola River by the Savannah. The portion of the paleo-Apalachicola between the present divide and the Savannah then reversed course. On their map, this distance is roughly 14km. Use of Hack's law converts this distance to a stream length of ca. 38km. Using 38km in Eq (1) yields a date of 126ka for the capture.

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

Figures

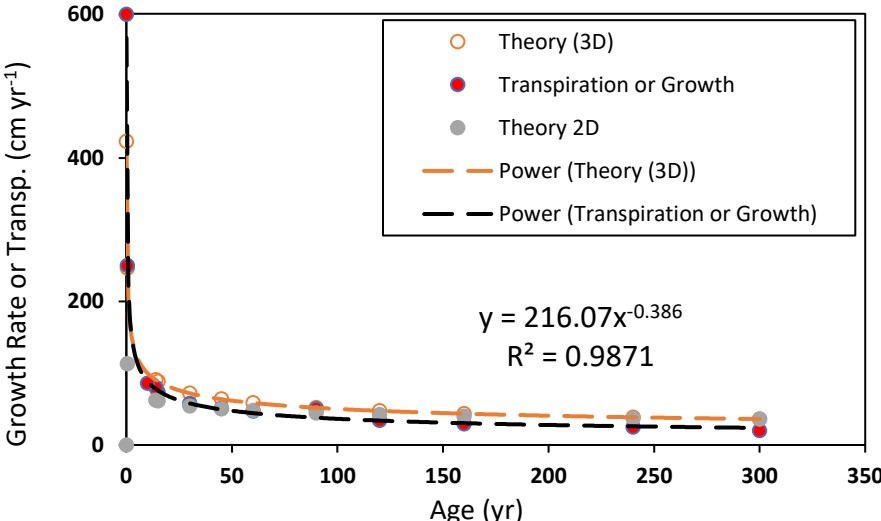

**Figure 1. Growth rates and transpiration rates of Eucalypts. The unknown parameter $x_0/t_0$ is estimated from the precipitation, which is variable in the study region, but typically a little over 2 meters/year where the trees were**

**found. The 3D value of the optimal paths exponent was in much closer accord with the data over the entire age range from 1 month to 300 years. The extracted parameters for transpiration and tree height as functions of time were indistinguishable (less than 1% different).**


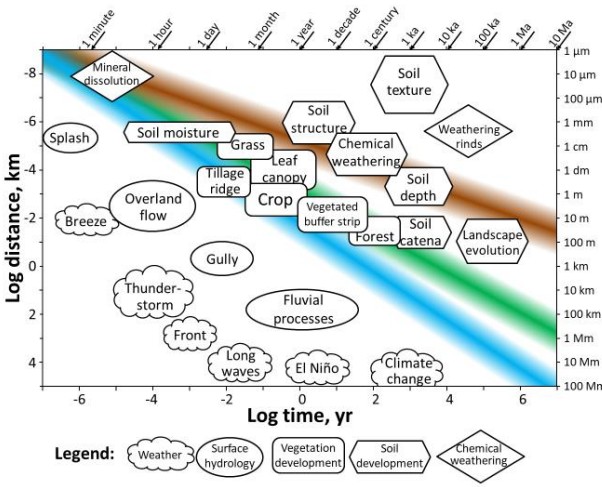


**Figure 2. Organization of hydrologic processes according to spatial and temporal scales with logarithmic axes (modified after Loague and Corwin (2006)). Thus, a constant slope on this figure represents the value of a power in a power-law. The blue strip represents water flow distances for scale-invariant flow rates a function of time and has slope 1. The green strip denotes root radial extent (and vegetation height) as a function of time and has slope 0.83. The brown strip represents soil depth as a function of**

**time and has slope 0.53. These slopes less than 1 imply that vegetation physical extent and soil depth have decreasing growth rates with time. The particular exponents that govern these scaling relationships are the optimal paths exponent in two dimensions for vegetation growth and the percolation backbone in three dimensions for soil formation. See the text for further detail.**

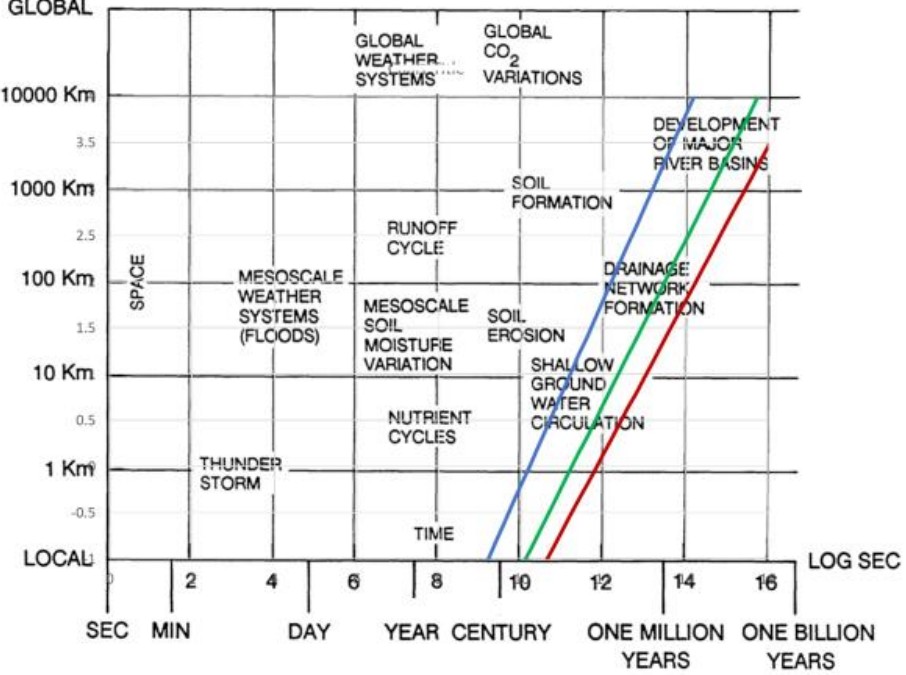


**Figure 3. Figure 2.9 from National Research Council (1991) overlain with predictions of the hierarchical growth of two-dimensional optimal flow paths in the subsurface (green and brown) and the associated prediction of the subsurface flow rate (in blue). The subsurface flow rate chosen was 2m yr⁻¹, equal to the geometric mean of the values used later in Figure 3 to show the typical range of annual flow rates. The green path uses the tortuosity exponent of standard percolation, 1.13 (applicable to more homogeneous models) and associated with a Hack's law exponent of 0.565, while the brown path uses the tortuosity exponent from the optimal flow paths in a highly heterogeneous environment, 1.21, which generates a Hack's law exponent of 0.605. The values 0.565 and 0.605 are shown in Hunt (2016) to give an excellent accounting for actual values of Hack's law relating stream length to drainage basin area (between 0.57 and 0.60) as reported in Maritan et al. (1996).**




**Figure 4 (after Hunt, 2017b). "Optimal paths" represents a prediction from scaling relationship Eq. 1 with s = 0.83 = 1/$D_{op}$t (2D) and a flow rate that ranges from 0.25myr⁻¹ to 25myr⁻¹. "Solute transport substitutes the backbone exponent, $D_b$ = for $D_{opt}$, but is otherwise unchanged. The flow rates relevant to vegetation growth and soil formation should be smaller than the range given in Bloeschl and Sivapalan (1995). Data corresponding to "optimal paths" include over 6000 results for tree heights (or root lateral spreads) as well as four river basin changes due to stream capture and, finally, the centroids of the icons for "Shallow groundwater circulation," "Drainage network formation," and "Development of major river basins" from NRC (1991). Data for "Solute Transport" correspond to soil depths from dozens of studies around the world, as well as new sources listed here, particularly commencing in the middle Mesozoic. Those with ages in the millions of years up typically describe depths of deep tropical weathering, or laterite soils. "Space limit," a maximum accessible scale is chosen as the linear dimension of a supercontinent, which we approximated as the square root of today's total continental area (ca. 12,500,000 km²), while "Time limit," presumed to be 150 Myr, is the length of time since the break-up of Pangaea (Cawood and Hawkesworth (2015). Thus, the limits are assumed imposed by the resetting of tectonic structures known as the Wilson cycle.**





**Figure 5.** Demonstrates that tectonic (constant velocity) and predicted optimal paths scaling, when using space and time scales defined at the pore scale and for pore-scale flow rates, converge at time scales larger than 1Myr and length scales exceeding 100km. In this, consider that 25m yr$^{-1}$ is less than a maximum flow rate (35m yr$^{-1}$ was also found here from data of Aggarwal et al. 2014) and that slower flow rates may also be relevant. For comparison see "Optimal (2.5m yr$^{-1}$)," which shows predictions of Eq. (1) with a flow rate of 2.5m yr$^{-1}$ instead of 25m yr$^{-1}$. If, as appears possible from the figure, drainage basin scaling results are found in the wedge of time and length scales between optimal path scaling and tectonic scaling, the range of drainage basin sizes becomes restricted more sharply due to limited water supply at shorter time scales than the values defined by the intersection of optimal paths with the Wilson cycle time scale. In particular, the time scale at this intersection for a flow rate of 1m yr$^{-1}$ and a tectonic rate of 0.035m yr$^{-1}$ is about 300kyr and the length scale is about 13km.

**Table 1. Comparison of Predicted and Actual River Lengths**

| River | Time (kyr) | Predicted Length (km) | Integrated Length (km) |
|---|---|---|---|
| Aare | 1300 | 266.3910909 | 450 |
| Rhine | 1200 | 249.3392696 | 300 |



| | | | |
|---|---|---|---|
| **Suarez** | **409** | **102.4381693** | **172** |
| **Daotang** | **80** | **26.59588435** | **15.453** |
| **Amazon** | **55000** | **5883.935575** | **6363** |
| **Gunnison** | **1000** | **214.4626345** | **290** |
| **Colorado** | **11000** | **1555.989809** | **2320** |
| **Orinoco** | **10000** | **1438.129244** | **840** |
| **Crab Creek drainage** | **5000** | **810.9849544** | **454.26** |
| **Crab Creek drainage** | **1500** | **299.8344999** | **57.75** |
| **Morocco** | **100** | **31.9819806** | **20** |
| **Morocco** | **100** | **31.9819806** | **40** |
| **Almanzora** | **1100** | **232.0387233** | **100** |
| **Niger** | **11000** | **1555.989809** | **4180** |
| **Ravi** | **15000** | **2010.61021** | **720** |
| **Sutlez** | **15000** | **2010.61021** | **1450** |
| **Chennab** | **15000** | **2010.61021** | **960** |
| **Jehlum** | **15000** | **2010.61021** | **725** |
| **Yangtze** | **95000** | **9243.446991** | **6300** |
| **Dadu** | **1000** | **646.4153097** | **500** |
| **Dadu** | **3800** | **283.216537** | **500** |
| **Blue Nile** | **24000** | **2964.982158** | **1450** |
| **Katonga** | **400** | **100.5716549** | **220** |
| **Meuse** | **650** | **150.2223584** | **150** |
| **Verde** | **2500** | **457.3278787** | **272** |
| **Salt** | **2500** | **457.3278787** | **320** |
| **Cahabón** | **155** | **44.71315568** | **50** |
| **Mojave** | **160** | **47.16279732** | **40** |
| **Mojave** | **3500** | **603.9411873** | **200** |
| **Yellow** | **8200** | **1220.589687** | **1300** |
| **Daotanghe** | **500** | **120.9390398** | **100** |

**Table 2. Predicted Spatial Scales at 150 Myr for Various Flow Rates**

| Flow Rate (m yr$^{-1}$) | Weathered Depth (m) | Drainage Basin Length (km) |
|---|---|---|
| 20 | 920 | 11,169 |
| 2 | 268 | 1665 |
| 0.2 | 78 | 248 |

725