# Peer review of "Non-linear Hydrologic Organization"

_Nonlinear Processes in Geophysics, 2021_

## Author Response (AR1)

1) General comment: I have found some difficulties in reading some passages in section 2.2 due to the very specific terminology used. For instance, terms like "sill overlapping" may have a very specific meaning in hydrology studies. However, because of the interdisciplinary character of NPG I suggest, if possible, to explain some of the specific terminology in such a way to make it more readable to a wider audience.

We have added discussion to both terms of "sill overtopping" and "headward erosion."

2) Figure 1) is plotted using linear axes. In order to better appreciate the power-law relation between the two related quantities, it would be better to plot it in in a log-log diagram. Indeed, the Pearson coefficient $R^2$ suggests that the scaling index is obtained by a linear regression fit on a log-log diagram.

We have made the change (including use of meters instead of centimeters).

3) At page 2 the Authors introduce two different dimensionality indices ($D_b$ and $D_{opt}$) without specifying how these dimension are estimated and/or the fact that these are related to the fractal dimension. I suggest the authors to spend some more words to explain the meaning of these dimensions and also to add some references to the percolation theory (perhaps the book by Stauffer and Aharony 2003, etc).

The exponents have been defined and their purpose outlined. We have added three references: two are to the specific exponent values found and their functions, while the other one is to Muhammad Sahimi's book, sometimes referred to as Stauffer II, and we have described the fundamental meanings and roles of these fractional exponents.

4) The terms in Eq. 2 are not clearly stated. I guess L is the stream length and A the drainage basin area. This should be clearly stated in the text so that the reader can immediately link the terms of Eq. 2).

The referee is correct and we now explicitly identify these definitions above the equation. This links into the next comment.

5) The exponent in Eq. 2) should be related to a sort of fractal dimension relating the link between area and length, i.e, indicating the anomalous scaling of the area as a function of the length. Some words regarding this issue should be included in the text.

We now also discuss the relationship between drainage basin length, defined in our manuscript as *l*, and drainage basin area, which is known to be Euclidean. This result provides a conceptual contrast to that of comment #4 and, we hope, give the necessary basis for understanding the proposed scaling relationship.

6) In the text the following unit measures are used: Ma and Myr (ka and kyr). I guess that both "a" and "yr" stand for years. If I am not wrong, and if there is not a specific reason to use "a" to indicate year I suggest to adopt a single notation. Otherwise, please explain the difference

between "a" and "yr". (Furthermore, check yr in the text because there are some typos where yr is written as y).

The referee is correct here, even though there is a distinction (ka means thousands of years before present, while kyr means thousands of years elapsed). We should adopt kyr (and Myr), since we are really interested primarily in time elapsed. However, when referring to dates rather than durations, this means that we need to use kyr "before present," for example, rather than just ka. However, we still agree with the referee, since the present venue is geophysical, not geological, where there would not be a confusion between the terms and their focus would be appreciated.

We have also replaced y by yr everywhere.

7) All the estimated quantities, such as the fit parameters reported in Figures 4 and 5, should include the errors. $R^2$ is not a sufficient qualifier for the goodness of the fit. Also parameter errors provides correct estimation of the uncertainties in the fitting procedure.

We neither fit any model to data shown in Figs. 4 and 5 nor report $R^2$. We only visually compare different scenarios and theoretical estimations with experimental observations. However, we now report average relative error and root mean square error for the predicted lengths reported in Table 1. We also conducted the t-test and found that the difference between the means of predicted and integrated values is not significant. A new statement is now added on page 10 to address the referee's comment.

8) The significance of $R^2$ when comparing different datasets, is not sufficient to estimate which one is the best. I suggest to make a significance test which takes into account the number of events in each set also.

Please see our response to the previous comment.

9) In the text the Authors uses cm, m, km as length measurements. I suggest to adopt the standard SI (International System) notation and its scheme.

Here, we agree with the referee to a point. We have eliminated cm, mm, etc. We find it, however, a bit unnatural to refer to continental length scales as $10^7$m and would like to retain 10,000km. This difficulty is only heightened by the occasional need to refer to areas, as $10^8$km$^2$ (or 100 million km$^2$) is much easier to process than $10^{14}$ m$^2$.

10) In the final section the Authors state that surface heterogeneities and non-linear dynamics. However, it is not clear what they exactly mean with the term non-linear dynamics or non-linear flow equations. The Authors should be more precise in defining the meaning and the equations they refer with the terms non-linear dynamics. What kind of dependence is it esxpectsd between the characteristic velocity and the increasing scale ? is it a linear or a non-linear dependence which may alter the expected scaling features ? Furthermore, they should better describe the interaction mechanism between surface and subsurface processes.

Ultimately, the non-linear dynamics usually refers back to the Navier-Stokes equation in some form. We now refer to the more general Kolmogorov's energy cascade and to a specific depth dependence of velocity in rivers and the "law of the wall." We note the fundamental role of surface roughness in atmospheric and fluvial processes. We also allow that hyporheic exchange of fluids between rivers and the subsurface may "muddy the waters" (not in so many words in the text). Much more could be said here, of course, but we hope that our brief description may suffice, since we only wish to compare fundamental results of our work with those of non-linear dynamics, rather than discuss non-linear dynamics in detail.

---

## Author Response (AR2)

We have completed the standard statements regarding acknowledgements, contributions, data, etc.